# Risk Stratification in Post-ERCP Pancreatitis: How Do Procedures, Patient Characteristics and Clinical Indicators Influence Outcomes?

**Kapil Kohli [1], Hrishikesh Samant [1], Kashif Khan [1], Sudha Pandit [1], Kelli Morgan [1], Urska Cvek [2], Phillip Kilgore [2], Marjan Trutschl [2], Eleni Mijalis [1], Paul Jordan [1], James Morris [1], Moheb Boktor [3] and Jonathan Steven Alexander [1,4,\*]**

1 Departments of Medicine/Section of Gastroenterology and Hepatology, Ochsner-LSU Health Sciences Center in Shreveport, Shreveport, LA 71103-3932, USA; kkohli@lsuhsc.edu (K.K.); Hsaman@lsuhsc.edu (H.S.); kkan@lsuhsc.edu (K.K.); spandit@lsuhsc.edu (S.P.); khuff@lsuhsc.edu (K.M.); emijal@lsuhsc.edu (E.M.); pjorda1@lsuhsc.edu (P.J.); jmorri2@lsuhsc.edu (J.M.)

2 Department of Computer Sciences, Louisiana State University-Shreveport, Shreveport, LA 71115, USA; urska.cvek@lsus.edu (U.C.); pkilgore@lsus.edu (P.K.); mtrutsch@lsus.edu (M.T.)

3 Department of Internal Medicine, Division of Digestive and Liver Diseases, University of Texas Southwestern Medical Center, Dallas, TX 75390, USA; moheb.boktor@utsw.edu

4 Department of Molecular and Cellular Physiology, Louisiana State University, 1501 Kings Highway, Shreveport, LA 71103-3932, USA

\* Correspondence: jalexa@lsuhsc.edu; Tel.: +1-318-675-4151

**Abstract:** Background. Post-endoscopic retrograde cholangiopancreatography (ERCP) pancreatitis (PEP) remains common, and severe complications are associated with ERCP. There is no previous study detailing the effect of race and gender in a US-based population on risk of PEP. Methods. Data were collected on 269 "first-performed" consecutive ERCPs followed by division by race (White vs. African-American) and sex (Female vs. Male). A total of 53 probable risk factors were evaluated by uni- and multivariate analysis followed by outcomes expressed as an odds ratio (OR) (with a 95% confidence interval, 95% CI). Finally, a principal component analysis was performed to construct a risk prediction model for PEP, which can be used by clinicians at bedside. Results. After analyzing the risk factors based on race and gender-based groups, Caucasian males with PEP are more likely to have prior history of pancreatitis ($p = 0.009$), lower hemoglobin ($p = 0.02$)/blood urea nitrogen (BUN) ($p = 0.01$)/creatinine before ERCP ($p = 0.07$) and lower BUN ($p = 0.01$)/creatinine after ERCP ($p = 0.07$), while Caucasian females with PEP are more likely to have higher white blood cell (WBC) count before ERCP ($p = 0.08$) and lower amylase ($p = 0.10$)/bilirubin ($p = 0.09$)/aspartate aminotransferase (AST) after ERCP ($p = 0.08$). African-American males with PEP are more likely to have lower weight ($p = 0.001$)/smaller height ($p = 0.0005$)/lower alkaline phosphatase ($p = 0.002$)/AST ($p = 0.04$)/alanine transaminase (ALT) ($p = 0.03$) before ERCP and lower alkaline phosphatase ($p = 0.002$)/AST ($p = 0.01$)/ALT ($p = 0.004$) after ERCP, while African-American females with PEP are more likely to have prior history of pancreatitis ($p = 0.004$)/higher lipase before ($p = 0.0001$) and after ($p = 0.05$) ERCP along with increased risk with pancreatic duct cannulation ($p = 0.0001$) and injection ($p = 0.0001$)/biliary sphincterotomy ($p = 0.0001$). Importantly, prior history of ERCP, elevated AST after ERCP, and BUN prior to ERCP were found to be important clinical features predicting post-ERCP pancreatitis. To our knowledge, this is a first known attempt at developing a risk scoring system for PEP in a US population with decision tree learning. Conclusions. It is very evident that both patient and procedure-related risk factors vary by race and gender in the US population, leading to the development of a new risk assessment tool for PEP that can be used in clinical practice. We need to follow up with a larger prospective study to validate this novel race and gender-based risk scoring system for PEP.

**Keywords:** ERCP; pancreatitis; lipase; ALT; AST; bilirubin

## 1. Introduction

Gastrointestinal endoscopy has been attempted for over 200 years, but the introduction of semi-rigid gastroscopes in the middle of the twentieth century marked the dawn of the modern endoscopic era. Since then, rapid advances in endoscopic technology have led to dramatic changes in the diagnosis and treatment of many digestive diseases.

Endoscopic retrograde cholangiopancreatography (ERCP) is an endoscopic technique in which a specialized side-viewing upper endoscope is guided through the mouth into the duodenum, the ampulla of Vater is identified, and it is cannulated with a thin plastic catheter, allowing for instruments to be passed into the bile and pancreatic ducts under fluoroscopic guidance. Its benefits—the minimally invasive management of biliary and pancreatic disorders—are challenged by a higher potential for serious complications than any other standard endoscopic technique [1].

Pancreatitis is the most common complication of diagnostic and therapeutic ERCP occurring in 1 to 15% of patients [2,3]. Pancreatic duct cannulation, rectal indomethacin, and hydration are well-known methods to reduce the incidence of post-endoscopic retrograde cholangiopancreatography (ERCP) pancreatitis (PEP). However, the prevalence of complications and procedure-related mortality does not appear to have decreased over time.

Although many studies have detailed the patient and procedure-related risks for the development of PEP, these reports suggest younger age, normal bilirubin, female gender, pancreatic duct injection, and biliary balloon sphincteroplasty as risks, but they have never been studied in context of race and gender to stratify risk, especially in United States (US) populations [2,4–6]. This is first study of a US population studying the effects of race and gender on PEP sought to predict risks for this complication, especially since ERCP is increasingly being performed on an outpatient basis, even for patients who require riskier procedures e.g., endoscopic sphincterotomy [6]. This can help decide who can be discharged to home safely after outpatient ERCP versus the need to admit for costly inpatient monitoring, thus helping reduce the cost of care in a world with increasing population and limited resources.

## 2. Materials and Methods

### 2.1. Study Population

This was a retrospective, single-center Institutional Review Board (IRB)-approved study conducted at the Louisiana State University Health Sciences Center in Shreveport (LSUHSC-S) on 509 consecutive ERCP procedures performed on 327 patients between 2011 and March 2017. Among these procedures, 27 were excluded from the study because of the age range being outside the study protocol (age of 18 to 75 years included in our study). After excluding all other racial groups, 269 of the first performed ERCPs performed in Caucasians and African-American patients were included in the final analysis. (Figure 1).

### 2.2. Study Protocol and Data Collection

This was a retrospective study for which de-identified information was collected after chart review so requirement for written informed consent was waived off. Data were collected regarding date of procedure, demographics, body mass index (BMI), smoking, and alcohol history, prior history of pancreatitis. Laboratory parameters that were collected before and after ERCP included complete blood counts (CBC), comprehensive metabolic panel (CMP), which includes 14 blood tests such as blood glucose, electrolytes, renal function tests, liver function test, calcium, amylase, and lipase, which were performed as part of standard of care evaluations (LSUHSC-S clinical laboratories.) Imaging studies including ultrasound of right upper quadrant, computed tomography (CT) of abdomen and pelvis, and magnetic resonance imaging (MRI)/Magnetic resonance cholangiopancreatography (MRCP) were also collected. Indication of ERCP procedure, findings during ERCP, difficulty of cannulation, pancreatic duct cannulation, pancreatic duct injection, pancreatic duct stent placement, balloon sphincterotomy, sphincteroplasty, and common bile duct (CBD) stenting were collected. Information about intravenous fluids given after ERCP and rectal

administration of indomethacin pre or post-ERCP was also collected. CBC, CMP, amylase, and lipase were collected before ERCP followed by second lab draw at 24 h.

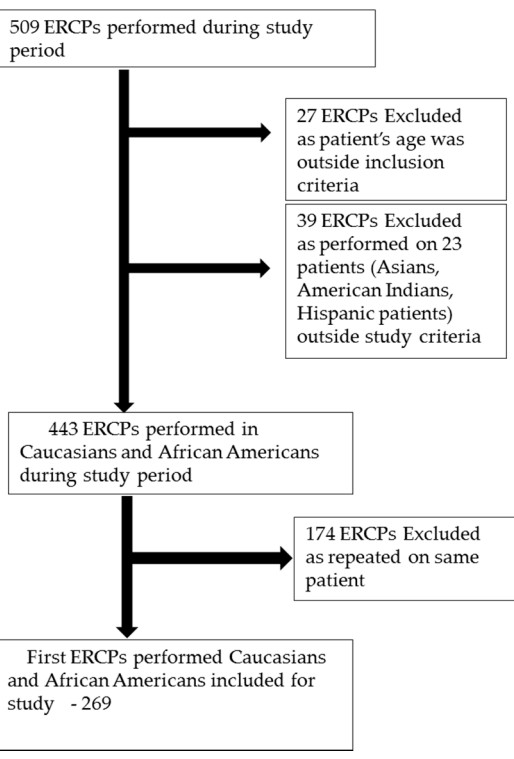

**Figure 1.** Study patient inclusion/enrollment.

*2.3. Definition*

A diagnosis of PEP was made when patients had a new onset or worsened pancreatic-type abdominal pain for >24 h after the procedure, with an increased serum amylase level more than thrice the upper limit of normal (ULN) [7].

*2.4. Statistical Analysis*

The data were tabulated with Microsoft Office software (Excel; Microsoft, Redmond, WA, USA) and analyzed using Graphpad Instat 3. A total of 53 potentially relevant risk factors were evaluated using univariate analysis with $\chi^2$ test. Variables with a P value less than 0.1 in the univariate analysis were included in the stepwise multivariate logistic regression analysis to identify the independent risk factors for PEP. ORs with 95% confidence interval (CI) were calculated. Goodness of fit for multivariate logistic regression model was evaluated using the Hosmer–Lemeshow test. The receiver operating characteristic (ROC) curve was plotted to reveal the cut-off values of optimal sensitivity and specificity. The area under the ROC curve (AUROC) was also calculated. The comparison of the occurrence rate of PEP was assessed using the $\chi^2$ test. A P value of less than 0.05 was considered statistically significant.

**3. Results**

Characteristics of patients undergoing ERCP procedures. Overall, 509 ERCP procedures were performed in 327 patients out of which only the first performed ERCPs in Caucasian and African-American patients were included in final analysis. In total, 23 patients (Hispanic, American Indian, and Asian patients) with 39 ERCP were excluded from final analysis, keeping in mind the objectives of the study. The subjects of first performed ERCPs included 55 (20.4%) Caucasian males, 83 (30.8%) Caucasian females, 40 (14.8%) AA males, and 91 (33.8%) AA females for a total of 269 ERCPs included in the study. PEP occurred in

22 (8.17%) of the first performed ERCPs. The indications for ERCP and techniques used in ERCP are shown in Tables 1 and 2.

**Table 1.** Indications for endoscopic retrograde cholangiopancreatography (ERCP).

| Indication for ERCP | *n* (%) |
|---|---|
| CBD Stone/Obstructive Jaundice | |
| | 195 (72.4%) |
| CBD Stricture | 20 (7.4%) |
| Cholangitis | 10 (3.7%) |
| Pancreatitis | 11 (4.0%) |
| Dilated CBD | 6 (2.2%) |
| Bile Leak | 20 (7.4%) |
| Unknown | 7 (2.6%) |

**Table 2.** Techniques used in ERCP.

| Techniques | *n* (%) |
|---|---|
| CBD Stone removal | 54 (20%) |
| CBD Stricture Dilatation | 29 (10.7%) |
| Endoscopic sphincteroplasty | 8 (2.9%) |
| Balloon Sweeps | 103 (38.2%) |
| CBD Stent placement | 154 (57.2%) |
| Endoscopic sphincterotomy | 139 (51.6%) |
| Pancreatic duct stent placement | 42 (15.6%) |

Cholelithiasis was the leading indication (72.4%), and the most common corresponding techniques were CBD stent placement (57.2%) and endoscopic sphincterotomy (51.6%). Pre- or Post-ERCP rectal indomethacin (100 mg of indomethacin rectally immediately before or after ERCP) was given in 141 (52.4%) of the total patients, while indomethacin was given to five (33.3%) patients with PEP. No deaths occurred directly due to PEP or other complications of ERCP.

Univariate analysis. In univariate analysis, 52 variables consisting of 35 patient-related factors and 17 procedure-related factors were assessed for PEP (Table 3).

**Table 3.** Univariate analysis of potential risk factors for post-endoscopic retrograde cholangiopancreatography (ERCP) pancreatitis (PEP).

| Variables Divided by Race and Gender | *n*/*N* (% or SD) or Value as Defined | *p* Value |
|---|---|---|
| Caucasian Males | | |
| Prior history of pancreatitis | PEP group = 6/8 (75%)<br>Non-PEP group = 9/46(19.5%) | 0.009 |
| Hemoglobin before ERCP | PEP group = 12.2 (SD = 1.76)<br>Non-PEP group = 12.8 (SD = 1.92) | 0.02 |
| BUN before ERCP | PEP group = 7.8 (SD = 4.12)<br>Non-PEP group = 13.6 (SD = 0.75) | 0.01 |
| Creatinine before ERCP | PEP group = 0.86 (SD = 0.18)<br>Non-PEP group = 1.17 (SD = 1.10) | 0.07 |
| BUN after ERCP | PEP group = 7.6 (SD = 3.15)<br>Non-PEP group = 13 (SD = 11.8) | 0.01 |
| Creatinine after ERCP | PEP group = 0.82 (SD = 0.23)<br>Non-PEP group = 1.19 (SD = 1.17) | 0.07 |

**Table 3.** *Cont.*

| Variables Divided by Race and Gender | n/N (% or SD) or Value as Defined | p Value |
|---|---|---|
| Caucasian Females | | p value |
| WBC count before ERCP | PEP group = 12.3 (SD = 3.43)<br>Non-PEP group = 8.66 (SD = 4.29) | 0.08 |
| Amylase after ERCP | PEP group = 87 (SD = 42.43)<br>Non-PEP group = 80.18 (SD = 83.35) | 0.10 |
| Bilirubin after ERCP | PEP group = 1.18 (SD = 1.04)<br>Non-PEP group = 2.23 (SD = 2.03) | 0.09 |
| AST after ERCP | PEP group = 47.60 (SD = 34.40)<br>Non-PEP group = 85.60 (SD = 96.76) | 0.08 |
| African-American Males | | p value |
| Weight in kilograms | PEP group = 70.8 (SD = 6.01)<br>Non-PEP group = 84.3 (SD = 20.18) | 0.0001 |
| Height in Inches | PEP group = 69 (SD = 4.35)<br>Non-PEP group = 70.5 (SD = 3.16) | 0.0005 |
| Alkaline phosphatase before ERCP | PEP group = 122 (SD = 64)<br>Non-PEP group = 423 (SD = 464) | 0.0002 |
| AST before ERCP | PEP group = 61 (SD = 42)<br>Non-PEP group = 152 (SD = 199) | 0.04 |
| ALT before ERCP | PEP group = 74 (SD = 46)<br>Non-PEP group = 171 (SD = 147) | 0.03 |
| Alkaline phosphatase after ERCP | PEP group = 131 (SD = 55)<br>Non-PEP group = 412 (SD = 422) | 0.002 |
| AST after ERCP | PEP group = 47 (SD = 29)<br>Non-PEP group = 125 (SD = 115) | 0.01 |
| ALT after ERCP | PEP group = 66 (SD = 29)<br>Non-PEP group = 171 (SD = 148) | 0.004 |
| African-American Females | | p value |
| Prior history of pancreatitis | PEP group = 4/6 (66.6%)<br>Non-PEP group = 9/85 (10.5%) | 0.004 |
| Lipase before ERCP | PEP group = 5620 (SD = 3006)<br>Non-PEP group = 1209 (SD = 5073) | 0.0001 |
| Lipase after ERCP | PEP group = 925 (SD = 1281)<br>Non-PEP group = 1462 (SD = 3852) | 0.05 |
| Risk with pancreatic duct cannulation | PEP group = 3/6 (50%)<br>Non-PEP group = 38/85 (45.8%) | 0.0001 |
| Risk with pancreatic duct injection | PEP group = 1/6 (16.6%)<br>Non-PEP group = 11/85 (12.9%) | 0.0001 |
| Risk with biliary sphincterotomy | PEP group = 5/6 (83.3%)<br>Non-PEP group = 68/85 (80%) | 0.0001 |

Significant patient-related risk factors included the following: Caucasian males with PEP are more likely to have prior history of pancreatitis ($p = 0.009$), lower hemoglobin ($p = 0.02$)/BUN ($p = 0.01$)/creatinine before ERCP ($p = 0.07$) and lower BUN ($p = 0.01$)/creatinine after ERCP ($p = 0.07$), Caucasian females with PEP are more likely to have higher WBC count before ERCP ($p = 0.08$) and lower amylase ($p = 0.10$)/bilirubin ($p = 0.09$)/AST after ERCP ($p = 0.08$), African-American males with PEP are more likely to have lower weight ($p = 0.001$)/smaller height ($p = 0.0005$)/lower alkaline phosphatase ($p = 0.002$)/AST ($p = 0.04$)/ALT ($p = 0.03$) before ERCP and lower alkaline phosphatase ($p = 0.002$)/ST ($p = 0.01$)/ALT ($p = 0.004$) after ERCP, African-American females with PEP are more likely to have prior history of pancreatitis ($p = 0.004$)/higher lipase before ($p = 0.0001$) and after ($p = 0.05$) ERCP. Among the potential procedural risk factors, three were found to be only associated to an increased risk of PEP in AA females: increased risk with pancreatic duct cannulation ($p = 0.0001$) and injection ($p = 0.0001$)/biliary sphincterotomy ($p = 0.0001$).

*Factor Analysis and Decision Tree Learning*

In order to gain a sense of which factors affected prediction in our model, we performed exploratory factors analysis (EFA) on the data. EFA was implemented using the R statistical programming language version 3.6.1 [8] and the principal() function from Revelle's psych package version 2.0.12 (Revelle, William. Psych: Procedures for Psychological, Psychometric, and Personality Research. Online: CRAN.R-project.org/package=psych). First, we prepared the data by removing any field from it that was (a) a subject identifier, (b) non-numeric, (c) which contained 50% or more missing values, or (d) had zero information entropy (e.g., was constant-valued). Nominal data were treated as numeric because they could be uniquely mapped to an integral value. In total, 18 out of 64 fields were removed. Any record that contained a missing value in the remaining columns was also removed, leaving 99 of 269 or 36.8% of the data. This was necessary because the EFA process does not admit these conditions existing. Then, we performed principal axis factoring (PAF) on the preprocessed data as implemented via the "psych" package for the R statistical language. This method is similar to principal component analysis but restricts the number of components to a given number of factors. We used the elbow method to determine the optimal factor count for EFA, which yielded an optimal factor count of 13 (Figure 2A).

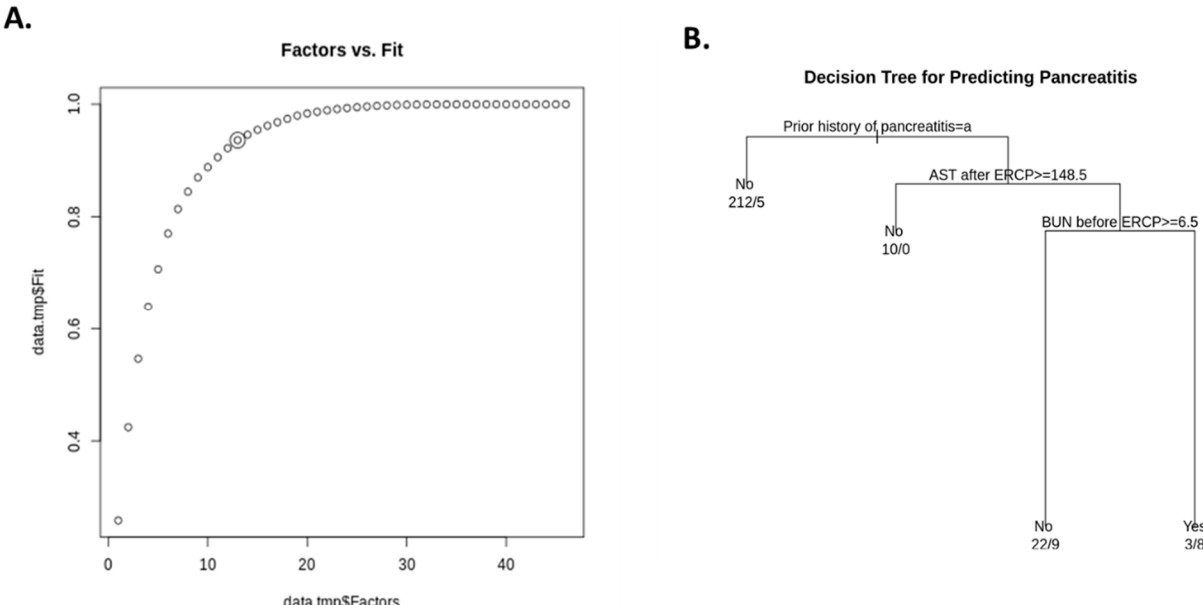

**Figure 2.** Elbow method extraction. The best factor (14) was extracted using the elbow method based on change in fit (**A**). (**B**) shows the clinical "decision tree" for predicting post-ERCP pancreatitis, which was created using Classification and Regression Trees (CART); major discriminative factors included prior history of ERCP, followed by AST after ERCP and BUN before ERCP.

We applied decision tree learning to these data using "rpart" version 4.1-15 package by Therneau and Atkinson (Therneau, Terry and Atkinson, Beth. "rpart: Recursive Partitioning and Regression Trees. Online: https://CRAN.R-project.org/package=rpart, accessed on 12 April 2019) for the R statistical programming language which is derived from the Classification and Regression Trees (CART) algorithm developed by Breimann et al. [9]. This classifier was trained to detect post-ERCP pancreatitis using 10-fold cross-validation (Figure 3A). Contributions of each dimension to the best PCA were identified as follows: (1) AST before ERCP, (2) platelets, and (3) patient weight, which were the factors found to contribute the most to the outcome. Data covering the entire range of false positives are missing, so the remaining range is linearly interpolated (shown as a dashed line). The true-positive rate (TPR) was found to increase dramatically with small increases with the false-positive rate (FPR), indicating good performance of the approach (the dotted line

depicts the performance of the classifier given random chance). Figure 2B shows the PCA with the best fit—most patients collected into a single cluster that failed to resolve into distinct subgroupings.

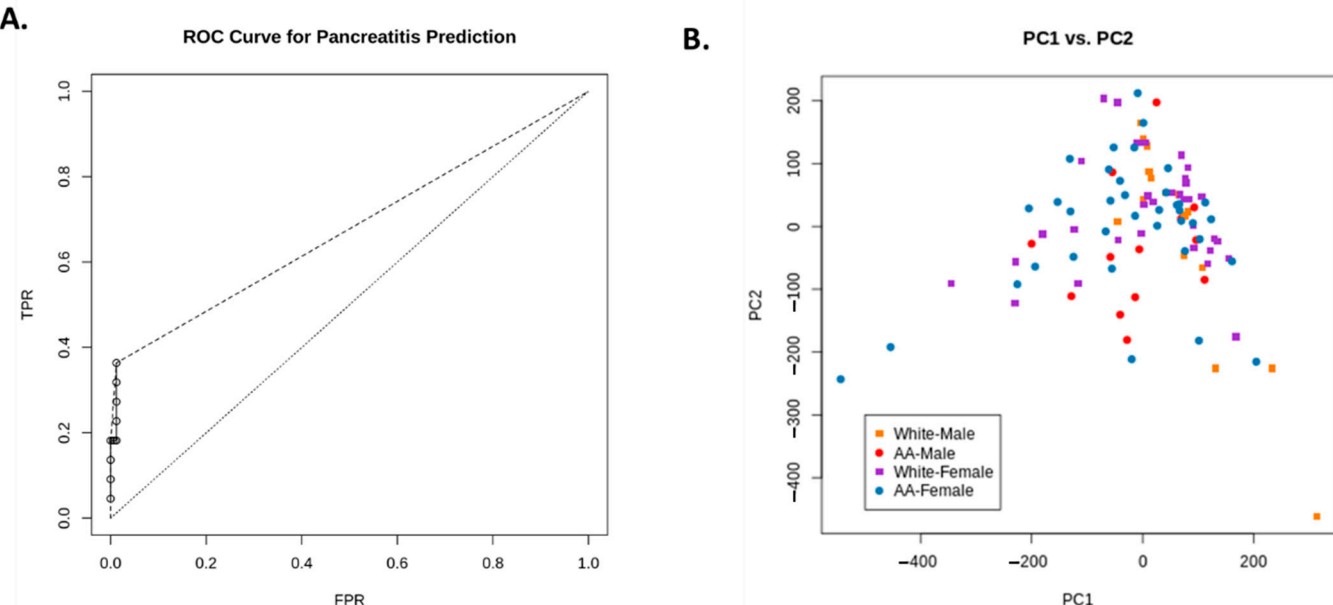

**Figure 3.** (**A**) shows the receiver operating characteristic (ROC) curve used to evaluate the performance of the classifier. Data covering the entire range of false positives is missing, so the remaining range is linearly interpolated (shown as a dashed line). The true-positive rate (TPR) increase dramatically with increases false-positive rate (FPR). The broken line shows the performance of the classifier given random chance. (**B**) shows the PCA with the best fit based on PC1 and PC2—most patients fell into a single cluster.

Lastly, Figure 3B shows the clinical "decision tree" for predicting post-ERCP pancreatitis, which was created using CART; the major discriminative factors identified included prior history of ERCP, followed by AST and BUN before ERCP.

## 4. Discussion

ERCP is a very important therapeutic procedure for pancreatic–biliary tract diseases. The incidence of post-ERCP pancreatitis has been estimated in several large clinical trials and ranges from 1.6 to 15%, with most studies demonstrating rates of 3 to 5% [2,4–6,10–19]. In our study, the occurrence of PEP was 8.17%, which is similar to ranges reported in previous studies [20–22].

A growing body of evidence has evaluated patient and procedure-related risk factors, helping us understand the mechanisms of post-ERCP complications Additional factors may influence the risk of PEP, but their roles have not been fully established [2,16]. Therefore, our study is different from other studies in two respects. First of all, several potentially significant risk factors, especially related to race and gender have been estimated for the first time for the US population. Secondly, a risk scoring system has been developed to predict the likelihood of developing PEP, which can be used by clinicians at bedside to decide on the need for extended monitoring.

We found that Caucasian males with PEP were more likely to have a prior history of pancreatitis ($p = 0.009$), lower hemoglobin ($p = 0.02$)/BUN ($p = 0.01$)/creatinine before ERCP ($p = 0.07$) and lower BUN ($p = 0.01$)/creatinine after ERCP ($p = 0.07$). By comparison, Caucasian females with PEP were more likely to have higher WBC counts before ERCP ($p = 0.08$) and lower amylase ($p = 0.10$)/bilirubin ($p = 0.09$)/AST after ERCP ($p = 0.08$).

African-American males with PEP are more likely to have lower weight ($p = 0.001$)/smaller height ($p = 0.0005$)/lower alkaline phosphatase ($p = 0.002$)/AST ($p = 0.04$)/ALT ($p = 0.03$) before ERCP and lower alkaline phosphatase ($p = 0.002$)/AST ($p = 0.01$)/ALT ($p = 0.004$)

after ERCP. African-American females with PEP are more likely to have prior history of pancreatitis ($p = 0.004$)/higher lipase before ($p = 0.0001$) and after ($p = 0.05$) ERCP. Among the potential procedural risk factors, three well-known factors were found to be only associated to an increased risk of PEP in AA females: increased risk with pancreatic duct cannulation ($p = 0.0001$) and injection ($p = 0.0001$)/biliary sphincterotomy ($p = 0.0001$).

Female gender stratified into Caucasian and AA females showed different apparent levels of risk factor for PEP in our study. Female patients undergoing ERCP procedures were seen to be at a higher risk of developing PEP in previous studies, but we found that AA females (6.3%) were less likely to develop pancreatitis compared to men (11.5%) [11,15,16,23,24]. This finding is in contradistinction from a previous study that shows that female gender and a young age are risk factors for pancreatitis [5].

A prior history of pancreatitis is considered to be a risk factor in the pathogenesis of PEP [25]. In our study, Caucasian males and AA females with a prior history of PEP had an increased risk of recurrence. Therefore, it is our recommendation that these patient subsets planning to undergo ERCP must be informed about the highly increased risk for PEP recrudescence along with the additional need to be monitored for a longer duration before discharge to minimize risk.

Procedure-related variables also play an important role in developing complications after ERCP, especially PEP. Among the potential procedural risk factors, only three were found to be associated with an increased risk of PEP in AA females: increased risk with pancreatic duct cannulation ($p = 0.0001$) and injection ($p = 0.0001$)/biliary sphincterotomy ($p = 0.0001$). Pancreatography was also identified as an independent risk factor in our present study, which is in line with findings from a prior report for similar risks performed in Japan [26].

Using our exploratory factor analysis method, we were able to select 13 clinical features, which were evaluated for their power in predicting PEP outcomes. Our classifier approach ultimately did not focus on any of these factors and instead focused primarily on whether or not patients had a prior history of pancreatitis, the levels of AST and BUN before ERCP. This approach might reflect most of the records not having a positive status for pancreatitis as well as the fact that there were comparatively few (22 versus 247) instances that did. It is probable that rpart interpreted this as the most effective way of explaining this phenomenon; indeed, removing this dimension causes rpart to conclude that all records should be classified as "no".

In the vast majority of cases, prior history of pancreatitis was indicative of whether or not patients would go on to develop pancreatitis; of the 217 individuals with no past history, only five had indicators for pancreatitis. Of those remaining individuals, a level of AST < 148.5 IU/mL and the level of BUN before ERCP < 6.5 mg/dL were found together to be important predictors of PEP. Therefore, in our present study, the co-presentation of these clinical indicators suggests an increasing tendency toward the development of post-ERCP pancreatitis, as is shown in Figure 2B.

We recognize several limitations of our study. First, the study was carried out only at a single tertiary referral care center located in Louisiana, which may have patients with lower socioeconomic status carrying increased overall health risk profiles e.g., comorbidities such as diabetes and hypertension which were not considered as variables, nor were family history and previous surgeries considered. Secondly, as a retrospective study, this analysis that might underestimate the occurrence of several undescribed complications. In addition, due to the retrospective nature and limited recorded data, we were not able to decide the exact timing of post-ERCP labs. For example, in our study, few important labs such as post-ERCP serum amylase levels did not reach the required level of significance. Furthermore, some known risk factors for PEP were not included, such as Sphincter of Oddi dysfunction and pancreatic sphincterotomy. Additionally, a large prospective study will be needed to evaluate this newly proposed scoring system.

## 5. Conclusions

In conclusion, several newly identified patient and procedure-related risk factors were found to be apparently involved in the development of PEP that need to be considered for outpatient ERCP to be stratified based on the need for extended monitoring before discharge. Additionally, to our knowledge, this is a first known attempt at developing a risk scoring system for PEP in the US population. Major medical society recommendations such as more consistent pre- or post-ERCP use of rectal indomethacin or appropriate pancreatic stent placement to prevent the development of PEP should be carefully considered in future studies.

**Author Contributions:** K.K. (Kapil Kohli) contributed to study concept and design, acquisition of data; analysis and interpretation of data, writing the manuscript for important intellectual content. H.S. contributed to study concept and design, acquisition of data; analysis and interpretation of data, critical revision of the manuscript for important intellectual content, material support; study supervision. K.K. (Kashif Khan) contributed to study concept and design, acquisition of data; analysis and interpretation of data. S.P. contributed to study concept and design, acquisition of data; analysis and interpretation of data, material support; study supervision. K.M. contributed to study concept and design, acquisition of data; analysis and interpretation of data. U.C. contributed to critical revision of the manuscript for important intellectual content; statistical analysis and technical data handling. P.K. contributed to critical revision of the manuscript for important intellectual content; statistical analysis, and technical data handling. M.T. contributed to critical revision of the manuscript for important intellectual content; statistical analysis. E.M. contributed to statistical analysis and technical data handling. P.J. contributed to study concept and design, material support; study supervision. J.M. contributed to study concept and design, material support; study supervision. J.S.A. contributed to study concept and design, critical revision of the manuscript for important intellectual content, study supervision, M.B. created IRB, contributed to study concept and design, critical revision of the manuscript for important intellectual content, study supervision. All authors have read and agreed to the published version of the manuscript.

**Funding:** This research received no external funding and was funded by the Department of Medicine, Division of Gastroenterology, LSUHSC-S.

**Institutional Review Board Statement:** The study was conducted according to the guidelines of the Declaration of Helsinki, and approved by the Institutional Review Board of LSUHSC-S "Effect of rectal Indomethacin in preventing post-ERCP pancreatitis: A retrospective analysis of epidemiology, patient characteristics, and management approach and post-ERCP outcome" (Moheb Boktor, MD, Principal Investigator, Protocol # STUDY00000591, approved 4-28-2016).

**Informed Consent Statement:** As this was a retrospective-study patient consent had been previously waived.

**Data Availability Statement:** Data available upon request.

**Conflicts of Interest:** The authors declare no conflict of interest.

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
