# Peer review of "Risk Stratification in Post-ERCP Pancreatitis: How Do Procedures, Patient Characteristics and Clinical Indicators Influence Outcomes?"

_pathophysiology, doi:10.3390/pathophysiology28010007_

Round 1
Reviewer 1 Report
This study's main objective is Risk Stratification in Post-ERCP Pancreatitis: How do Procedures, Patient Characteristics, and Clinical Indicators Influence Outcomes? The experimental work appears to have been carried out well. However, a few points deserve attention for further publication. I suggest that it is accepted for publication after the following revisions:
INTRODUCTION: What is the novelty of the work when compared to other works of literature? This information must be evident in the introduction.
DISCUSSION: Can the results achieved be compared with other works in the literature?
CONCLUSIONS: The main contributions to the accomplishment of this work must be included in the conclusion. Please, authors must use numbers.
- Please check all references according to the author's instructions.
- Include more details in the figures (error bars) and table captions.
- The manuscript must be formatted according to the journal's standards.
Author Response
Thanks for the thoughtful comments and recommendations
- What is the novelty of the work when compared to other works of literature? To our knowledge, this is the first known attempt at developing a risk scoring system for PEP in US population with decision tree analysis. Introduction edited
- Can the results achieved be compared with other works in the literature? - Yes. We do suggest large prospective studies to validate in the conclusion
-
Please check all references according to the author's instructions--Checked
- Include more details in the figures (error bars) and table captions--edited.
- The manuscript must be formatted according to the journal's standards--checked.
Reviewer 2 Report
This manuscript focus on a popular and trendy topic.
However, there are several points that rise questions:
There are not many studies concerning race, but gender is well studied in this topic - PEP risk factors.
The sentence "Despite...over time." in lines 52-54 page 2, is incorrect because the studies about prophylactic methods such as indomethacin, pancreatic duct stenting, and intensive hydration confirmed an effect in PEP prevalence, lowering its risk.
As far as I understand, one of the goals of the study is tho define a model with risk factors that can help clinicians to decide whether to discharge or not patients. To do that, it is required a must larger group of patients (bigger N) because of the importance of that decision on patients´safety. A lower (or normal) amylase on the first 6 h is a good predictor, and that is not discussed in the manuscript.
Methods and statystics are unclear and disorganized. Why were other races excluded and only african americans studied? If the race is potentially related to a higher or lower risk, so every race should be tested.
The risk factors present after the ERCP event are not usefull. However, if the goal is to use them as an indicator of likelihood to discharge a patient with lower risk, then they should be studied in a different way.
Indomethacin administration can act as an important bias. Because it was given in only 52.4% of cases, (half of N) it can produce a bias. Was the indomethacin uptake bigger in some of the studied sub groups? (caucasian VS AA males, etc. )
Author Response
1) The sentence "Despite...over time." in lines 52-54 page 2, is incorrect because the studies about prophylactic methods such as indomethacin, pancreatic duct stenting, and intensive hydration confirmed an effect in PEP prevalence, lowering its risk.
-Completely agree with comments. We did change the context in the manuscript. Despite, prevalence of PEP has not changed significantly is worth noting.
2) As far as I understand, one of the goals of the study is tho define a model with risk factors that can help clinicians to decide whether to discharge or not patients. To do that, it is required a must larger group of patients (bigger N) because of the importance of that decision on patients´safety. A lower (or normal) amylase on the first 6 h is a good predictor, and that is not discussed in the manuscript.
-Yes we suggest large number prospective study. We did expect that low amylase is favorable factor. However, due to retrospective nature, we were not able to decide the timing of amylase levels and it did not reach statistical significance. We address this as limitation of study
3) Methods and statistics are unclear and disorganized. Why were other races excluded and only african americans studied? If the race is potentially related to a higher or lower risk, so every race should be tested.
-We agree that all races are important. We excluded them as number was less than 2. In prospective studies this factors can be considered
4) The risk factors present after the ERCP event are not use full. However, if the goal is to use them as an indicator of likelihood to discharge a patient with lower risk, then they should be studied in a different way.
We agree Post ERCP labs do not carry equal weight as pre ERCP labs once PEP sets inn diagnosis. We did study them to know their effect on monitoring patients and to include them for statistical scoring with decision tree analysis as we found significant difference in few of them during analysis. Our priority in the study was racial difference though
5) Indomethacin administration can act as an important bias. Because it was given in only 52.4% of cases, (half of N) it can produce a bias. Was the indomethacin uptake bigger in some of the studied sub groups? (Caucasian VS AA males, etc. )
Thank you for very good review. We did thought of use of Indomethacin as bias. On Revision we found that Indomethacin use was completely related to performing physician assessment risk of PEP with no relation to any subgroups. We also did not find any statistical difference
Reviewer 3 Report
In this study, authors analyzed the risk factors of post-ERCP pancreatitis and attempted to create a risk scoring system based on their data from a single, tertiary center in the US. The topic is important, I would have some questions and some suggestions regarding the paper.
- Line 52 “Despite technological progress and recommendations of scientific societies, the incidence of complications and procedure-related mortality does not appear to have decreased over time.”
- Line 55 “Although many studies have detailed the patient and procedure-related risks for development of PEP, these reports suggest younger age, normal bilirubin, female gender, pancreatic duct injection and biliary balloon sphincteroplasty as risks, but have never been studied in the context of race and gender to stratify risk, especially in a U.S. populations”Please provide references for the above statements.
- Line 58 “This first study of U.S population…” After “this” “is” missing.
- line 61: “e.g., endoscopic sphincterotomy4”, 4 should be indexed as a ref.
- line 66 IRB should be spelt out
- line 69 I wonder why the authors chose this age range?
- line 78 “Labs”: laboratory parameters/values should be used in my opinion
- line 79 CMP is not evident for every reader, please detail the parameters included in the panel at first mention
- line 83 IV should be spelt out it is the only mentioning of this
- line 84 Rectal administration of indomethacin
- line 84 Indomethacin should not be capitalized
- line 87 the definition of PEP is slightly different from consensus paper definition by Cotton et al 1991. Why? If you used that it should be referenced here.
- Statistical analysis section: software mentioned here should be correctly referenced (manufacturer, state, version etc.
- Table 2 I find it interesting that EST was less common than stent placement in first ERCP cases. In our practice, this is not so common not to perform sphincterotomy in these patients. Did you put in stents without sphincterotomy? How did you remove the stones without EST (195 patients with jaundice and stones)?
- Prophylactic pancreatic duct stents should also be mentions and analyzed as a proven effective method in PEP prophylaxis. In which cases were PPS inserted?
- line 144 here R packages are mentioned which I do not see in the Methods section, please mention it there (which program was used, which package for which tests, models.).
- Not so widely used statistical methods should also be referenced.
- Line 174 ERCP is not really a diagnostic tool anymore, only in rare cases.
- Line 176 references should be indexed
- Line 204 How could these patients have prior PEP? These were first ERCPs.
- Line 224 and 225 BUN and AST units are missing
Author Response
1) Line 52 “Despite technological progress and recommendations of scientific societies, the incidence of complications and procedure-related mortality does not appear to have decreased over time.”
-Changed and edited in manuscript
2)Line 55 “Although many studies have detailed the patient and procedure-related risks for development of PEP, these reports suggest younger age, normal bilirubin, female gender, pancreatic duct injection and biliary balloon sphincteroplasty as risks, but have never been studied in the context of race and gender to stratify risk, especially in a U.S. populations ”Please provide references for the above statements.
-added
- Line 58 “This first study of U.S population…” After “this” “is” missing.
- line 61: “e.g., endoscopic sphincterotomy4”, 4 should be indexed as a ref.
- line 66 IRB should be spelt out
- line 69 I wonder why the authors chose this age range?
- line 78 “Labs”: laboratory parameters/values should be used in my opinion
- line 79 CMP is not evident for every reader, please detail the parameters included in the panel at first mention
- line 83 IV should be spelt out it is the only mentioning of this
- line 84 Rectal administration of indomethacin
- line 84 Indomethacin should not be capitalized All suggested changes done. Thanks for the review-
- line 87 the definition of PEP is slightly different from consensus paper definition by Cotton et al 1991. Why? If you used that it should be referenced here--- Reference added
- Statistical analysis section: software mentioned here should be correctly referenced (manufacturer, state, version etc.--added the details, reference provided
- Table 2 I find it interesting that EST was less common than stent placement in first ERCP cases. In our practice, this is not so common not to perform sphincterotomy in these patients. Did you put in stents without sphincterotomy? How did you remove the stones without EST (195 patients with jaundice and stones)?--EST was done in all cases with CBD stone disease. No reported EST with stent placement may be at discretion of performing gastroenterologists due to sludge only and my be due to retrospective data collection
- Prophylactic pancreatic duct stents should also be mentions and analyzed as a proven effective method in PEP prophylaxis. In which cases were PPS inserted?--15% received stenting but did find significant difference in reducing PEP
- line 144 here R packages are mentioned which I do not see in the Methods section, please mention it there (which program was used, which package for which tests, models.).--reference provided
- Not so widely used statistical methods should also be referenced.--reference provided 26
- Line 174 ERCP is not really a diagnostic tool anymore, only in rare cases.--sentence changed in manuscript
- Line 176 references should be indexed-added
- Line 204 How could these patients have prior PEP? These were first ERCPs.--it was pancreatitis history and not PEP. Typo. Thanks for informing
- Line 224 and 225 BUN and AST units are missing--added